# Foreground-attention in neural decoding: Guiding Loop-Enc-Dec to reconstruct visual stimulus images from fMRI

## Abstract

The reconstruction of visual stimulus images from functional Magnetic Resonance Imaging (fMRI) has received extensive attention in recent years, which provides a possibility to interpret the human brain. Due to the high-dimensional and high-noise characteristics of fMRI data, how to extract stable, reliable and useful information from fMRI data for image reconstruction has become a challenging problem. Inspired by the mechanism of human visual attention, in this paper, we propose a novel method of reconstructing visual stimulus images, which first decodes the distribution of visual attention from fMRI, and then reconstructs the visual images guided by visual attention. We define visual attention as foreground attention (F-attention). Because the human brain is strongly wound into sulci and gyri, some spatially adjacent voxels are not connected in practice. Therefore, it is necessary to consider the global information when decoding fMRI, so we introduce the self-attention module for capturing global information into the process of decoding F-attention. In addition, in order to obtain more loss constraints in the training process of encoder-decoder, we also propose a new training strategy called Loop-Enc-Dec. The experimental results show that the F-attention decoder decodes the visual attention from fMRI successfully, and the Loop-Enc-Dec guided by F-attention can also well reconstruct the visual stimulus images.

## 1 Introduction

In recent years, reconstructing visual stimulus images from fMRI has gradually gained attention, which provides the possibility of "mind reading" in the future (Fig. 1). Existing work has shown that there is a certain mapping relationship between visual stimuli and brain activity (Poldrack & Farah, 2015), which provides us with the possibility and basis for reconstructing visual stimuli from fMRI data. The fMRI data which is collected from the human brain records the variations in blood oxygen level dependent (BOLD) and reflects the activity of nerves in human brain. Through the analysis of fMRI data, the correlation between brain activity and visual tasks can be explored, helping us to understand human visual mechanism better.

**Prior studies.** In recent years, there has been a lot of research in this field. The main methods can be roughly divided into the following two categories: **Linear regression models**, which encode fMRI data into image pixel values, and finally achieve image reconstruction. **Deep learning models**, such as DCNN and GAN model frameworks.

For the first type of methods, Yoichi Miyawaki et al. applied a multi-scale linear weighting model to predict the pixel value of each image and obtain the reconstruction results of the black and white simple images (Miyawaki et al., 2008). Fujiwara et al. proposed to use Bayesian canonical correlation analysis (BCCA) to build a reconstruction model, which extracts image information from measured data and reconstruct images (Fujiwara et al., 2013). Marcel AJ van Gerve et al. adopted a hierarchical generative model composed of Boltzmann machines with restricted conditions, which explores the reconstruction of feature hierarchies based on learning (Van Gerven et al., 2010). Yu et al. advanced a correlation network framework that could be flexibly combined with diverse pattern representation models. They revealed the effective connectivity of human brain and reconstructed the visual stimulus images (Yu & Zheng, 2017; Yu et al., 2018).

For the second type of methods, Guohua Shen et al. proposed an end-to-end deep convolution model to reconstruct visual stimulus images directly by using fMRI data, and at the same time they used GAN framework to optimize the training of the model (Shen et al., 2019a). In response to the current problem of the lack of data sets used to reconstruct stimulus images from fMRI data, Roman Beliy et al. suggest an approach for self-supervised training on unlabeled fMRI data (Beliy et al., 2019). Yunfeng Lin et al. proposed a model called DCNN-GAN by combining a reconstruction network and a GAN module. They utilized the CNN for hierarchical feature extraction and the DCNN-GAN to reconstruct more realistic images (Lin et al., 2019). Tao Fang et al. defined a Shape-Semantic GAN, considering the functional differences of different visual cortex areas, and reconstruct visual stimulus images under the guidance of shape and semantic information (Fang et al., 2020). The ideas of DCNN and GAN have also been applied to various other model frameworks to achieve the reconstruction of visual stimulus images (Seeliger et al., 2018; VanRullen & Reddy, 2019; Zhang et al., 2020; Qiao et al., 2020; Mozafari et al., 2020).

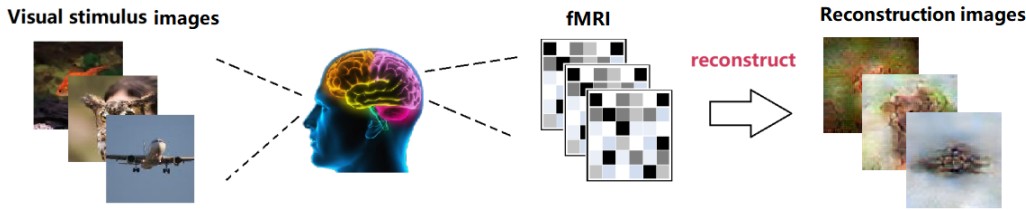

Figure 1: The work in this paper. The purpose of this article is to extract useful information from the visual cortex of the human brain, and build a model to reconstruct visual stimulus images from fMRI data by using the uesful information.

**Our contributions.** Because of the complex neural information in the visual cortex, how to extract valuable information from high-dimensional fMRI data has been a challenging problem. Inspired by human visual attention mechanism, we introduce visual attention to this work for the first time. We first decode the human visual attention from a small amount of fMRI data successfully, and then use visual attention to guide the work of reconstructing visual stimulus images from fMRI. Existing studies have shown that the human visual system is more inclined to focus on prominent objects, and the neural representation of these prominent objects in the brain is more obvious (Ungerleider & G, 2000; Braun et al., 2001). The distribution of attention leads to an information bottleneck, only the most prominent objects are allowed to appear in the inferior temporal cortex, especially the ventral visual stream that encodes the identity of the object. The visual attention mechanism is crucial for simulating the neural response of the higher visual system (Poggio & Anselmi, 2016; Khosla et al., 2020). Based on the above conclusions, we propose Foreground-attention (F-attention). We abstract the distribution of human visual attention as the most prominent foreground object distribution area in the images. Through the constructed F-attention decoder, the fMRI corresponding to the natural stimulus images can be decoded into visual attention distribution (Fig. 2a).

Because human brain is strongly wound into sulci and gyri, some spatially adjacent voxels are not directly connected in the human brain (Tallinen et al., 2014), the correlation between voxels is not only determined by location. Therefore, we need to consider global information when decoding fMRI, so we first introduce the self-attention module (Vaswani et al., 2017; Wang et al., 2018; Zhang et al., 2019) into the fMRI decoding process, the self-attention module can capture global information of the input data and expand the receptive field (Fig. 3c).

At the same time, we propose a new encoding and decoding framework called Loop-Enc-Dec (Fig. 2b,c). The first step we pre-train the fMRI encoder on the data set to realize the encoding process from images to fMRI. The second step we train the end-to-end encoder-decoder model under the guidance of F-attention, and input the images decoded by the decoder into the fMRI encoder for encoding to add the re-encoding constraint in the loss function, that is called "Loop-Enc-Dec". In the training process, natural pictures without fMRI data are also added for end-to-end self-enc-dec training, which increases the stability and generalization of the model. After experimental evaluation, the performance of Loop-Enc-Dec under the guidance of F-attention is better than that of mainstream methods. The main contributions of this article are as follows:

- As far as we know, we first introduced the visual attention mechanism to the work of reconstructing visual stimulus images from fMRI. According to neuroscience research, we abstract visual attention as the attention distribution of foreground objects (F-attention), and build a visual attention decoder to decode the human visual attention distribution from fMRI successfully.

- To our best knowledge, we first introduced the self-attention module into the work of fMRI decoding to capture global information of fMRI data.

- We propose a new enc-dec framework called Loop-Enc-Dec, guided by F-attention, which adds more loss constraints to the training of the image reconstruction model. Evaluating the quality of reconstructed images, our method outperforms previous works.

## 2 METHODS

In this section, we first introduce the data set used in the experiments, then we introduce the details of F-attention and Loop-Enc-Dec framework proposed in this article, as well as the step-by-step training strategy for the model. Finally, our evaluation method is given.

### 2.1 FMRI DATA SET

We use a publicly available benchmark data sets (Horikawa & Kamitani, 2017), which is widely used in the work of visual stimulus image reconstruction. In the image presentation experiment, fMRI signals were measured while subjects viewed a sequence of object images from ImageNet (Deng et al., 2009). The image presentation experiment consisted of two sessions: the training image session and the testing image session. In the training image session, 1200 images from 150 object categories (8 images from each category) were each presented once. In the testing image session, 50 images from 50 object categories (one image from each category) were each presented 35 times. The data collectors performed their analysis for each combination of feature types/layers and brain regions of interest (ROIs; V1–V4, the lateral occipital complex (LOC), fusiform face area (FFA), parahippocampal place area (PPA), lower visual cortex (LVC; V1–V3), higher visual cortex (HVC; covering regions around LOC, FFA and PPA) and the entire visual cortex (VC; covering all of the visual subareas listed above), the voxel size is 3×3×3 $mm^3$) (Horikawa & Kamitani, 2017).

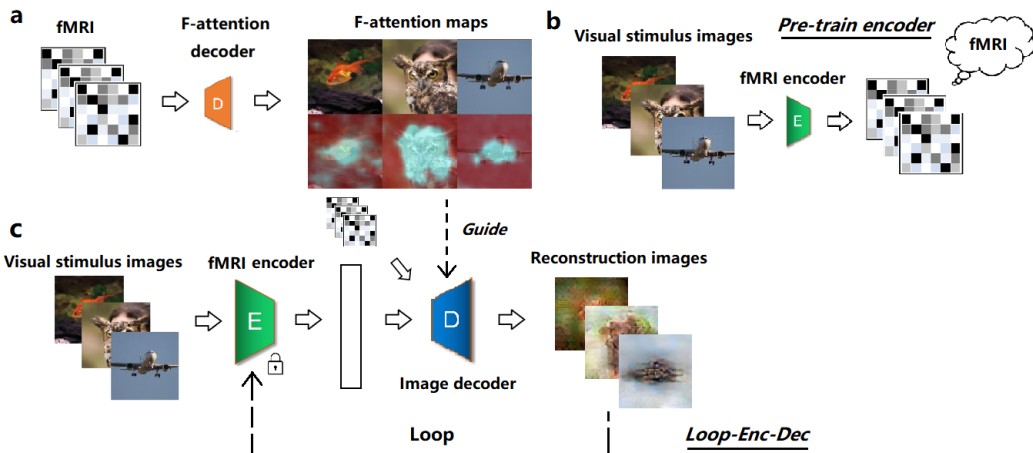

Figure 2: The overall model framework of the paper. (a) The overall structure of F-attention decoder. (b)(c) The Loop-Enc-Dec. We first pre-train a fMRI encoder to encode images to fMRI, and then use an image decoder guided by F-attention to reconstruct images, during the training, decoded images loop back to fMRI encoder.

## 2.2 PROPOSED MODEL

**Summarizing the proposed model.** We first trian the F-attention decoder on fMRI data(Fig. 2a), then pre-train the image encoder(Fig. 2b), next we fix the weight of the encoder, train the decoder under the guidance of F-attention(Fig. 2c). When testing, we first use fMRI as the input of F-attention decoder to decode the distribution of F-attention, then input the fMRI into the image decoder to reconstruct the visual stimulus images guided by corresponding F-attention.

### 2.2.1 F-ATTENTION

In order to get more useful information from fMRI data to guide the work of visual stimulus image reconstruction, we introduced the human visual attention mechanism into the work. We abstracted the human visual attention as foreground attention (F-attention) based on neuroscience research (Poggio & Anselmi, 2016; Khosla et al., 2020). By constructing a F-attention decoder, we decode human visual attention distribution from fMRI data.

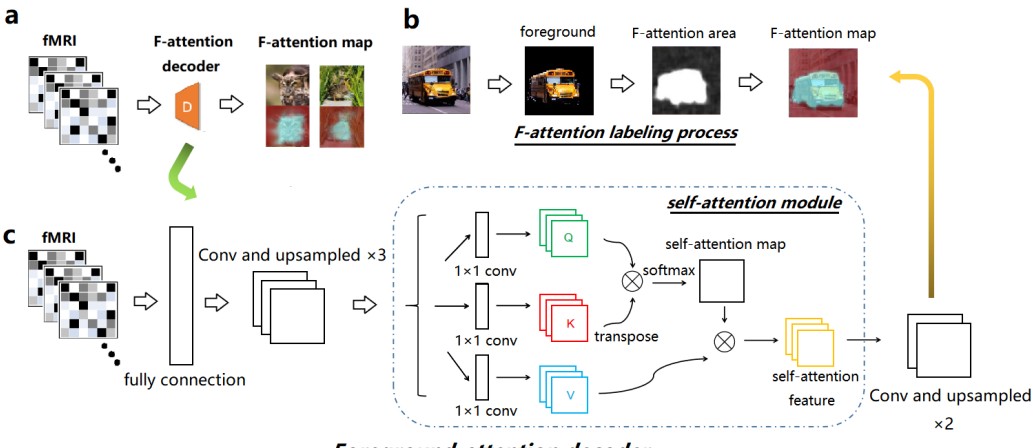

Figure 3: Foreground-attention. (a)The overall structure of F-attention decoder, through the F-attention decoder, we can decode the human visual attention distribution from the visual neural representation of the brain. (b)The human visual attention distribution labeling process of training set. (c)The detailed structure of F-attention decoder.

**Visual attention labeling.** For the purpose of obtaining the human visual attention distribution of the training set, we first label the train data (Fig. 3b). According to the definition of F-attention, we first extract the prominent objects in the natural images, and then perform the binary value on the images. Through the transformation operation, the main distribution areas of F-attention are obtained. Then we assigned attention weights to different regions in the images (the bluer the color, the higher the attention weights, in the train set, the weights of the foreground object areas are 1, the background areas are 0), the final F-attention maps are obtained as the human visual attention labels of the training set.

**F-attention decoder.** Then we construct F-attention-decoder to realize the decoding process from fMRI data to human visual attention(Fig. 3c). First, the preprocessed fMRI data is integrated into a vector, then the vector is sequentially connected to a fully connection module, convolution and upsampling modules, a self-attention module, convolution and upsampling modules, and we finally get the F-attention map. The self-attention module calculation formula is as follows, $d_k$ is 1:

$$Self - Attention(Q, K, V) = softmax(\frac{QK^T}{\sqrt{d_k}})V \quad (1)$$

In the training process, we use the mean square error loss (MSE) to constrain. After 400 rounds of iterations, the error decreases and stabilizes. The loss function is as follows:

$$Loss_{F-attention}(FA_i, FA_i^*) \propto \sum (FA_i - FA_i^*)^2 \quad (2)$$

Where $FA_i$ represents the visual attention distribution corresponding to the images in the training set, and $FA_i^*$ stands for the visual attention distribution decoded by the F-attention decoder.

### 2.2.2 LOOP-ENC-DEC

The Loop-Enc-Dec reconstruction model we proposed is divided into two steps for training, pre-training encoder (Fig. 2b) and training Loop-Enc-Decoder guided by F-attention (Fig. 2c).

**Pre-train fMRI encoder.** The fMRI encoder realizes the encoding process from natural images to fMRI. The structure of the encoder is mainly composed of residual network, convolution and full connection. The front end of the encoder is used as ResNet50 network (He et al., 2016) pre-trained on the ImageNet dataset, this part can extract image features (Fig. 4). The loss function is composed of the weighted sum of the mean square error (MSE) and the cosine similarity, as shown below:

$$Loss_{fMRI} \propto \alpha \sum (x_i - x_i^*)^2 + \beta \sum cos\left(\angle (x_i, x_i^*)\right) \tag{3}$$

Where $\alpha$ and $\beta$ are hyperparameters, $x_i$ is the original fMRI label, and $x_i^*$ is the encoding result. In this paper, $\alpha$ is assigned a value of 0.85 and $\beta$ is -0.15. This loss function is also used in the training process of Loop-Enc-Dec.

**Train Loop-Enc-Decoder guided by F-attention.** We fix the weight of pre-trained fMRI encoder, then connect an image decoder to form an end-to-end enc-dec model. The so-called 'Loop' idea is to re-input the reconstructed images into the encoder for cyclic encoding in the process of training Enc-Dec, increasing the fMRI encoding loss constraint, so that the original end-to-end model is transformed into a loop structure. The image decoder is composed of full connection, convolution and upsampling. At the same time, F-attention is used as a weight to multiply the convolution feature to guide the training process of the model (Fig. 4).

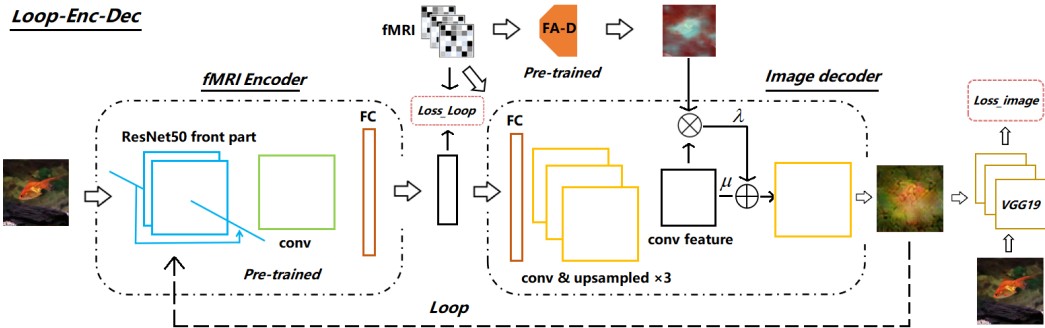

Figure 4: Detailed structure of Loop-Enc-Dec. We fix the network parameters of the pre-trained encoder, and train the Image decoder under the guidance of F-attention to realize the work of visual stimulus image reconstruction.

The F-attention ($FA$) decoded by pre-training F-attention decoder and the convolution features ($ConvF$) are used as a Hadamard product, the output is weighted $\lambda$ and added to the original input convolution feature weighted $\mu$. Then we complete the guidance of the attention weight, as shown in the following formula:

$$Feature_{FA} = \lambda FA \circ ConvF + \mu ConvF \tag{4}$$

In the training process, $\lambda$ is assigned a value of 0.7 and $\mu$ is 0.3. We also add unlabeled ImageNet images for end-to-end unsupervised self-enc-dec to enhance the generalization ability of the model. The loss function in the training process consists of three parts, as follows:

$$Loss_{LoopEncDec} = Loss_{image} + Loss_{Loop} + Loss_{imageNet} \tag{5}$$

Next, we will introduce the three components of $Loss_{LoopEncDec}$ in detail.

1) $Loss_{image}$. During training, as shown in Fig. 4, the image decoder has two inputs, one is the original fMRI data, we input it into image decoder to generate image $img_1$, and the other is the

output of pre-trained fMRI encoder(encoding the visual stimulation image), we input it into image decoder to decode image $img_2$. Then as shown in Fig. 4, we use $VGG19$ (Simonyan & Zisserman, 2014) as a feature extractor to extract the convolution features of visual stimulation images and reconstructed images ($img_1$, $img_2$), then calculate the loss between convolution features as shown in Eq. 6.

$$Loss_{image} \propto \sum [V(img_1) - V(img)]^2 + \sum [V(img_2) - V(img)]^2 \qquad (6)$$

Where $V$ represents the feature extractor composed of pre-trained $VGG19$.

2) $Loss_{Loop}$. Then we loop $img_1, img_2$ back into the fMRI encoder to encode two different features, which we names $fea_1, fea_2$. We use $Loss_{fMRI}$ to calculate the loss between encoded features ($fea_1, fea_2$) and raw fRMI data, as shown in Eq. 7.

$$Loss_{Loop} = Loss_{fMRI}(fea_1, x) + Loss_{fMRI}(fea_2, x) \qquad (7)$$

Where $x$ represents the raw fRMI. $Loss_{fMRI}$ is defined in Eq. 3.

3) $Loss_{imageNet}$. In order to enhance the generalization and robustness of the model, we introduce the image data without fMRI label from imageNet in the training process. Firstly, the image is input into the fMRI encoder for encoding, and then the encoding result is input into the image decoder for decoding to obtain the reconstructed image $img_3$. We also calculate the loss between $VGG19$ convolution features.

$$Loss_{imageNet} \propto \sum [V(img_3) - V(imgNet)]^2 \qquad (8)$$

Where $Loss_{fMRI}$ is Eq. 3, and $imgNet$ is the imageNet image without fMRI labels.

## 2.3 EVALUATION METHOD

This paper uses a quantitative method to evaluate the reconstructed visual stimulus images. The evaluation method is called pairwise similarity comparison (Shen et al., 2019b; Fang et al., 2020), we use 2-way method used in Fang et al. (2020). There are a total of 50 natural stimulus images in the test set. For each reconstructed image, it is calculated SSIM (Wang et al., 2004) value separately from the 50 natural stimulus images. We define the SSIM between the reconstructed image and the corresponding original image as $SSIM_{pair}$, and the SSIM with the remaining 49 as $SSIM_{random}$, then $SSIM_{pair}$ and $SSIM_{random}$ compare the SSIM values in pairs, a total of 50×49=2450 comparisons, we count the correct of times ($SSIM_{pair}$ is greater than $SSIM_{random}$), and calculate the correct rate(Shen et al., 2019b). We will supplement the details of this part in the appendix.

## 3 RESULTS

In the following, we will present the F-attention decoding results, visual stimulus image reconstruction results, some ablation experiments involving the model, and verify some conclusions in neuroscience. All our experiments are performed on the data set Horikawa2017 (Horikawa & Kamitani, 2017). There are a total of 1200 images in the training set and a total of 50 images in the test set. There is no overlap in image categories between the training set and the test set.

### 3.1 F-ATTENTION DECODEING RESULTS

We use entire visual cortex (VC) of fMRI data labeled with visual attention distribution for training. A total of 280 rounds of experiments are performed. The learning rate is set by using the SGDR method (Loshchilov & Hutter, 2016). According to SGDR, we divide the 280 rounds of training into four small rounds. The learning rate of each small round rises to the maximum value of 0.03 through a linear function, and then decays to the minimum value of 4e-5 through cosine annealing. Each small round repeats this operation to realize the periodic restart of the learning rate and improve the convergence speed and generalization of the model, the details are shown in the appendix. The results of decoding the F-attention distribution from fMRI are shown in Fig. 5, the more blue the area, the higher the weight of visual attention. From the results, it can be seen that the F-attention decoder we design can decode the distribution of visual attention successfully, whether the object is an animal, a man-made object, or a human, such as the fourth image in the second row, the duck in the grass can be accurately found, the seventh image in the fourth row, visual attention distribution area of the person lying on the hammock can also be accurately found.

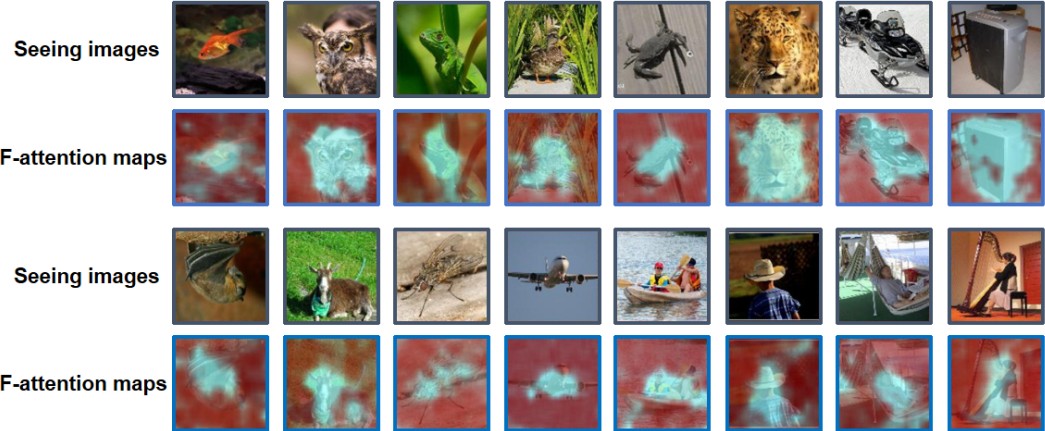

Figure 5: F-attention map decoding results of test data set. The more blue the area, the higher the weight of visual attention. The distribution of F-attention in the images can be accurately found.

## 3.2 VISUAL STIMULUS IMAGES RECONSTRUCTION RESULTS

In this part, we present the reconstruction results of our method and compare the results with other existing works (Shen et al., 2019a; Beliy et al., 2019; Fang et al., 2020; Mozafari et al., 2020), the comparison results are shown in Fig. 6. Through the evaluation method of pairwise similarity comparison mentioned above, we quantitatively evaluate the results of the experiment. It can be seen from the experimental results that our method can effectively reconstruct the images seen by humans from fMRI, and better decode the shape and direction of the prominent object. Compared with the previous work (reconstruction results of the visual stimulus images shown in Fang et al. (2020)), some improvements have been obtained. From a subjective perspective, our results can better pay attention to the distribution of foreground objects in the images. For example, for the reconstruction of the first goldfish image, the position and color of the goldfish in the image are more accurate. From a quantitative comparison point of view, our method experiment results achieve 68.9% by pairwise SSIM comparison, slightly better than 68.4% (Beliy et al., 2019), and better than 66.8% (Fang et al., 2020), 63.8% (Shen et al., 2019a) and 54.3% (Mozafari et al., 2020), indicating the effectiveness of our method and foreground-attention can guide loop-Enc-Dec to complete the reconstruction of visual stimulation images successfully.

## 3.3 MODEL ABLATION EXPERIMENTS

Here, we conduct ablation experiments to discuss the role of F-attention and Loop-Enc-Dec in reconstruction of visual stimulus images, the results are shown in Fig. 7. First, we give the results obtained using the complete method of this paper (the second row of Fig. 7), the pairwise SSIM value is 68.9%. Then on the basis of the complete method, the F-attention guidance is removed from the training of the decoder, and only the Loop-Enc-Dec structure is used for step-by-step training. First, the fMRI encoder is pre-trained, then the Loop-Enc-Dec is trained. We finally use the image decoder to test and get the reconstruction results (the third row of Fig. 7), the pairwise SSIM value is 66.7%. Then we remove the $Loss_{Loop}$ on the basis of the complete method, just use the image loss of end-to-end encoding, and at the same time train the image decoder under the guidance of F-attention and obtain the reconstruction results (the fourth row of Fig. 7), The value of pairwise SSIM is 65.0%. Finally, on the basis of the complete method, we remove the F-attention guidance and the $Loss_{Loop}$, and only use the end-to-end enc-dec for step-by-step training. The image reconstruction results are shown in the fifth row of Fig. 7, the pairwise SSIM value is 62.5%.

Comparing the ablation experimental results, it is not difficult to see that our proposed F-attention and $Loss_{Loop}$ have played a positive role in the image reconstruction work. Comparing the results in the second row with the results in the fifth row, it can be seen that under the combined effect of F-attention and $Loss_{Loop}$, the image reconstruction quality has been greatly improved, including

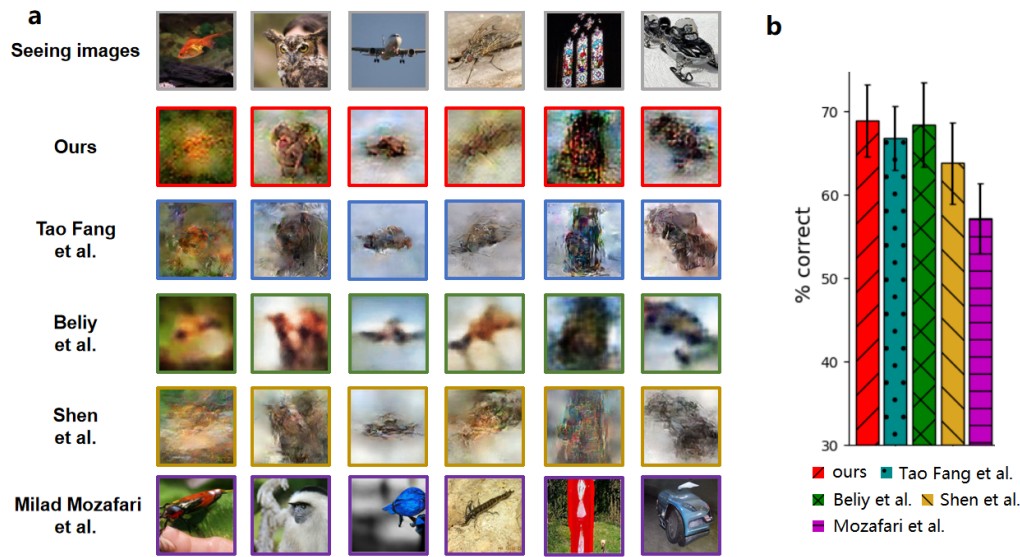

Figure 6: (a) Partial reconstruction results compared with other methods. (b) Result comparison with pairwise similarity, As mentioned in Fang et al. (2020), the quantitative evaluation value is the average result of five experiments

the shapes and colors of the prominent objects in the images, 68.9% is better than 62.5% in pairwise SSIM quantification.

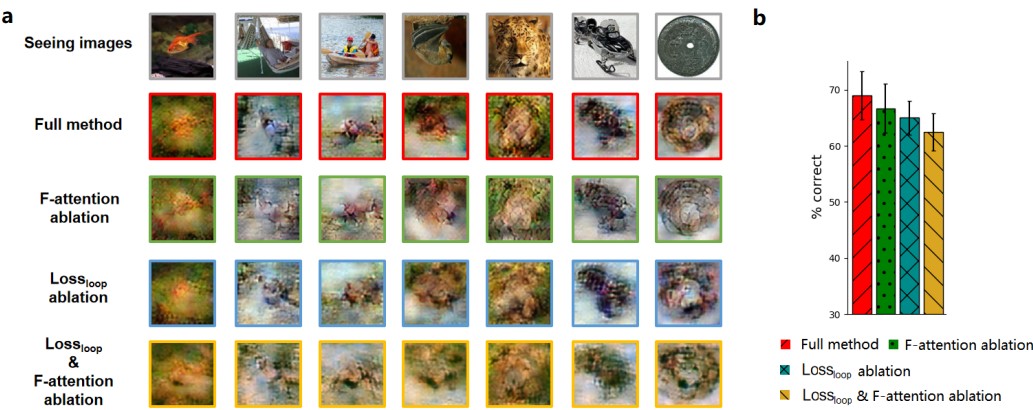

Figure 7: Model ablation experiments. (a) Visual stimulus images reconstructed by full method, full method without F-attention, full method without $Loss_{loop}$, full method without F-attention and $Loss_{Loop}$. (b) Comparing ablation results by pairwise SSIM.

**Comparing the ablation experimental results separately.** Comparing the results of the second row and the third row, it can be seen that F-attention plays a certain guiding role in the model, making the reconstruction of foreground objects in the images more prominent, such as the goldfish in the water in the first column, and the people lying in a hammock in the second column, pairwise SSIM quantitative comparison shows that 68.9% is better than 66.7%. The guiding effect of F-attention can also be obtained by comparing the results of fourth rows and fifth rows. The reconstruction results of foreground objects of fourth row are more prominent, and pairwise SSIM quantitative comparison shows that 65.0% is better than 62.5%. Comparing the results of the second row and the fourth row, it can be seen that the $Loss_{Loop}$ has made the details of the reconstruction images more clear and accurate. For example, the face of the leopard in the fifth column is clearer, and the shape

and details of stone pan are more accurate in the seventh column. The pairwise SSIM quantitative comparison shows that 68.9% is better than 65.0%. The constraint effect of $Loss_{Loop}$ can also be obtained by comparing the results of the third row and the fifth row. The details and quality of the image reconstruction in the third row are better. The pairwise SSIM quantitative comparison shows that 66.7% is better than 62.5%. According to the above ablation experiments, we proved the guiding role of F-attention in image reconstruction and the constraint role of $Loss_{Loop}$ in the training process.

### 3.4 VALIDATION OF NEUROSCIENCE RESEARCH

We verify some results of neuroscience research through experiments in the following. It is generally believed that in the human visual cortex, the low-level visual (LVC) cortex often pays attention to the shape and contour information of objects, and has significant directional sensitivity, while the high-level visual cortex (HVC) may correspond to the category information, movement information and other higher-level information of objects (Carandini et al., 2005; DiCarlo et al., 2012; Bracci et al., 2017; Zeman et al., 2020). We reconstruct the visual stimulus images with the fMRI data of the low-level visual cortex and the high-level visual cortex respectively. The results are shown in the Fig. 8, the pairwise SSIM value of the images reconstructed from LVC is 64.7%, and HVC is 55.7%. By comparing the experimental results, we can find that the LVC reconstruction results largely restore the shape and direction information of the foreground objects in the images, while the HVC reconstruction results lose the shape and direction information of the foreground objects. Quantitatively, the pairwise SSIM value of LVC is also better than HVC. This results verify that low-level visual cortex (LVC) pays more attention to the shape and contour information of objects than high-level visual cortex (HVC).

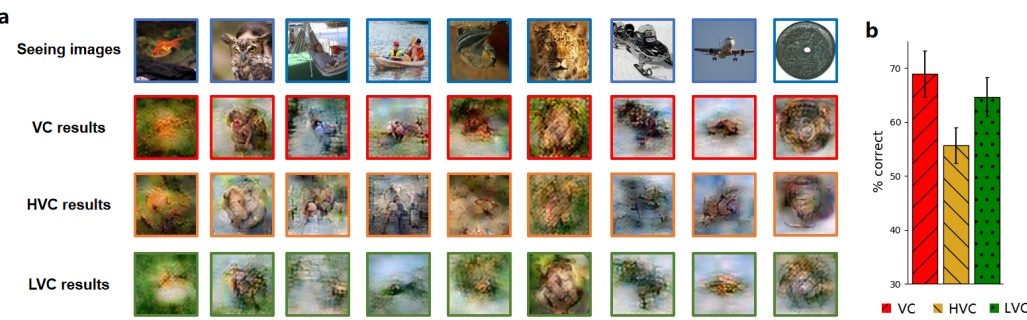

Figure 8: Validation of neuroscience research. (a) Visual stimulus images reconstructed from different visual cortex areas, including VC, HVC and LVC. (b) Result comparison with pairwise SSIM.

## 4 CONCLUSION

In this paper, inspired by human visual attention, we define the Foreground-attention distribution of people when they view natural images, and successfully decode the visual attention distribution (Foreground-attention) from the fMRI data of human brain. We achieve the decoding and interpretation of the visual attention of the human brain. The results show that areas with higher attention are concentrated on prominent objects in the images, such as animals and animal faces in the images, while lower attention is given to messy background information. This result is in line with the research conclusions on human brain visual attention (Ungerleider & G, 2000; Braun et al., 2001; Poggio & Anselmi, 2016; Khosla et al., 2020). At the same time, we propose a new enc-dec training strategy called Loop-Enc-Dec, and apply the decoded F-attention to guide it, then reconstruct the visual stimulus images from the fMRI data successfully. Compared with the previous work, the reconstructed visual images in this paper are more realistic and get a higher score on pairwise SSIM. From the perspective of neuroscience, we verify the following conclusions through experiments: The low-level visual cortex pays more attention to the shape of objects in the scene and has significant directionality, while the high-level visual cortex is insensitive to shape and direction information.

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

## A  APPENDIX

### A.1  DETAILS OF MODELS AND TRAINING

In this paper, the details of the F-attention decoder we designed are as follows, the input fMRI data is a one-dimensional vector, and the fully connected layer is connected, the output size is 512*10*10, then reshape it into the feature of size (10, 10, 512). The convolutions has step size of 2 and a kernel size of 4. The dimension of Query and Key is 68*68*8, and the dimension of Value is 68*68*64. In Loop-Enc-Dec, the front-end ResNet50 feature extractor in the fMRI decoder selects 'conv3-block4-out' as the feature output, the output feature size is 14*14*512, followed by two layers of residual networks, one layer of max-pooling and convolution (the size of the convolution kernel is 3, the step size is 1, padding is 'same'), and finally outputs the encoded results through a layer of fully connection. In order to eliminate the checkerboard effect that appears in the reconstructed image, in image decoder, we set the size of the convolution kernel to be twice the convolution step length, and use linear interpolation for each upsampling to expand the feature to four times the original.

### A.2  CHANGES IN LEARNING RATE USING SGDR

The linear function and cosine annealing function used in each round are as follows:

$$lr_{SGDR} = \begin{cases} \frac{(E_{cur}+1)*l_{max}}{E_w+1}, E_{cur} < E_w \\ l_{min} + \frac{1}{2}(l_{max} - l_{min}) * (1 + cos(\frac{E_{cur}-E_w}{E_{per}-E_w})), E_{cur} \geq E_{warm} \end{cases} \tag{9}$$

$E_{cur}$ counts how many epochs have been performed since the last restart, $E_w$ represents the 'warm' epochs of each small round , $E_{per}$ represents the total epochs of each small round, $l_{max}$ represents the maximum learning rate, and $l_{min}$ represents the termination learning rate.

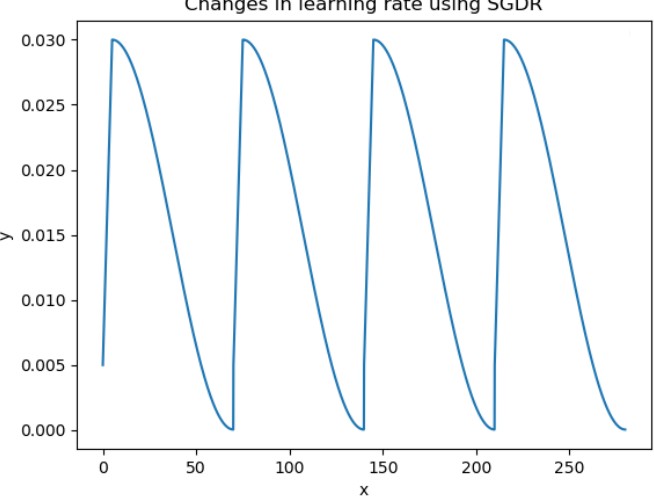

Figure 9: Changes in learning rate using SGDR

A total of 280 rounds of experiments were performed. The learning rate is set using the SGDR method (Loshchilov & Hutter, 2016). According to SGDR, we divide the 280 rounds of training into four small rounds. The learning rate of each small round rises to the maximum value of 0.03 through a linear function, and then decays to the minimum value of 4e-5 through cosine annealing. Each small round repeats this operation to realize the periodic restart of the learning rate and improve the convergence speed and generalization of the model.

Table 1: Determining the value of $\lambda$ and $\mu$ in Eq.4

| $\lambda$ | 0.1 | 0.3 | 0.5 | 0.7 | 0.9 |
|---|---|---|---|---|---|
| $\mu$ | 0.9 | 0.7 | 0.5 | 0.3 | 0.1 |
| $PairSSIM$ | 0.660 | 0.676 | 0.671 | 0.689 | 0.681 |

A.3   THE DETAILS OF SSIM

SSIM is an evaluation index to measure the similarity of two images, the calculation formula is as follows:

$$SSIM = \frac{(2\mu_x\mu_y + c_1)(2\sigma_{xy} + c_2)}{(\mu_x^2 + \mu_y^2 + c_1)(\sigma_x^2 + \sigma_y^2 + c_2)} \tag{10}$$

Where $x$ and $y$ are two images, $\mu_x$ is the mean of $x$, $\mu_y$ is the mean of $y$, $\sigma_x^2$ is the variance of $x$, $\sigma_y^2$ is the variance of $y$, $\sigma_{xy}$ is the covariance of $x$ and $y$, $c_1 = (k_1 L)^2$, $c2 = (k_2 L)^2$, is a constant used to maintain stability. $L$ is the dynamic range of pixel values. $k_1$=0.01, $k_2$=0.03.

A.4   HOW TO DETERMINE THE VALUE OF $\lambda$ AND $\mu$ IN EQ.4

We tested the sequential combination parameters of $\mu$ = 0.1, 0.3, 0.5, 0.7, 0.9 and $\lambda$ = 0.9, 0.7, 0.5, 0.3 and 0.1 respectively, and then selected the best combination, $\lambda$ = 0.7 and $\mu$ = 0.3, as shown in Table 1. It may not be the best among all the values of $\lambda$ and $\mu$, but it is appropriate.

A.5   CHANGE OF LOSS FUNCTION DURING TRAINING

In the process of training, the loss function Eq. 5 of each part can converge well. As shown in Fig. 10, figure a) shows the downward trend of total loss $Loss_{Loop-Enc-Dec}$ in the training process, figure b) shows the downward trend of $Loss_{Loop}$, figure c) and d) show the change trend of $Loss_{image}$, and figure e) shows the change trend of $Loss_{imageNet}$. The overall trend of error decreases and can be stabilized to a certain level. The cyclical rise and fall is due to the using of SGDR learning rate change strategy.

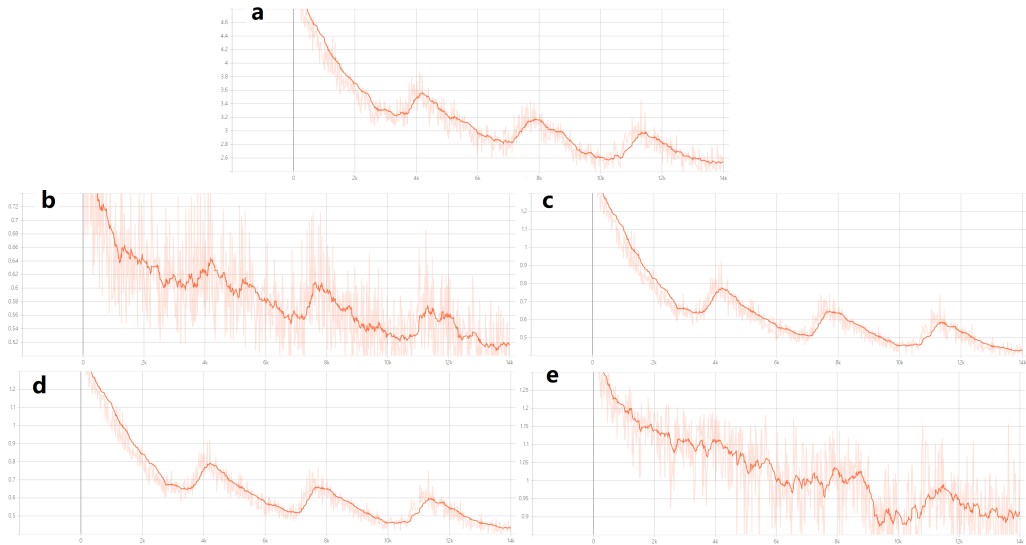

Figure 10: Change of loss function during training

