# OpenReview forum: "Foreground-attention in neural decoding: Guiding Loop-Enc-Dec to reconstruct visual stimulus images from fMRI"
_ICLR.cc/2022/Conference — ICLR 2022 Submitted_

### Official Review · Reviewer_48tB · 2021-11-01

**Correctness:** 2
**Technical Novelty And Significance:** 2
**Empirical Novelty And Significance:** 3
**Recommendation:** 5
**Confidence:** 5

**Main Review:**

My main concern is that, (if I understand the training process correctly) the label information used for training the attention decoder is from another attention map generator (the F-attention labeling process), which produces artificial attention map. Then what is the meaning of fitting the BOLD signal to the artifical attention map (see Fig.3a&b)?  Obviously this kind of fitting can give an attention map based on the fMRI data, which is similar to the F-attention map given in Fig.3b after optimization. However, this may not reflect what the human attention looks like in this task.

Other concerns:
1) The description of 'Visual attention labelling' should be made more clearer, e.g., how to extract the prominent objects in the images, is it a DNN-based objected detection algorithm? What does "perform a binary value" mean? What is a "tranform operation"?
2) The MSE loss in high dimension may not be easy to optimize (Eq.3, also in Eq.4). Could you please provide some demonstration of the training loss process, and give more illustration of how far the optimization goes?
3) The authors should make more clearer of the contribution of each part in their proposed model. Even though each part in explained in Sec.2, it is still very hard to know the underlying reason of such design. In addition, the ablation study does not make much sense. The increase of performance can be explained by including more training parameters.
4) The choose of  $\alpha$ and $\beta$ in Eq.4 (also $\lambda$ and $\mu$ in Eq.5) seems to be empirical.
5) The example of the decoded goldfish is not representative, because the decoded position of other images are also very accurate, as showed in Fig.6.
6) The argument in Sec.3.4 is wrong. Firstly, the authors should define clearly about LVC and HVC. Otherwise it is hard to say which one is more responsible for the shape perception and contour perception. At least, it shouldn't be in primary visual cortex.
7) What is $d_k$ in Eq.2?

Suggestions:
1, The F-attention is the key in this work, but also a strong hypothesis assumed by the authors that human will adopt to do image classification in such fMRI task. However, the authors should be careful when use this term, as this kind of simple task may not require attention in the human brain. On the other hand, even the brain activation (at the temporal lobe) may reflect the attention info, it is unclear which kind of attention will be involved, only bottom-up attention, or a mixture of top-down feedback and bottom-up attention [1]? So it is better to use more accurate terms, such as 'visual saliency map [2]', etc.
2, To get the attention labels, it is better to use the eye tracking data which is a more direct reflection of the attention of participants, rather than relying on the algorithm-based attention labeling. Such kind of data (3T fMRI + eye tracking data) may be available on the website.
3, The terms in Eq.7&8&9 should be showed in the figure to help understanding.

[1] Gilbert, C. D., & Li, W. (2013). Top-down influences on visual processing. Nature Reviews Neuroscience, 14(5), 350-363.

[2] Li, Z. (2002). A saliency map in primary visual cortex. Trends in cognitive sciences, 6(1), 9-16.


**Summary Of The Paper:**

The authors proposed a model to decoding the fMRI signal from the human visual cortex by introducing the Foreground-attention. They also proposed a enc-dec training strategy called Loop-Enc-Dec, which is guided by the F-attention, to successfully reconstruct the visual images from the fMRI data. A higher score based on pairwise SSIM is achieved compared to previous works.

**Summary Of The Review:**

The idea using F-attention to guide the network training is fine from the view of network training, but the way of using it to do image reconstruction from the FMRI data seems incorrect.  Fitting the F-attention model to the artifical attention map does not make much sense.

---

> ### Author Response · Authors · 2021-11-21
> **To Reviewer 48tB**
>
> **5) The example of the decoded goldfish is not representative.**
>
> Through comparison, it can be found that the color and shape of our goldfish reconstruction results are more accurate. In the comparison work Fang et al.[1], only 7 reconstruction results shown in the paper are presented, and the reconstruction results of the other 43 images are missing. Therefore, we can't visually compare the reconstruction results of the other 43 images.
>
> **6) The argument in Sec.3.4 is wrong.**
>
> Firstly, we define HVC as high-level visual cortex and LVC as low-level visual cortex in the paper. These partitions are common and have been divided in data preprocessing. Secondly, we intuitively compared the HVC and LVC reconstruction results, and found that the LVC reconstruction results can better retain the distribution area and approximate shape of significant objects in the image. This finding is also reflected in the quantitative evaluation. Therefore, we speculate that this may be because the low-level visual cortex can better capture the shape information of objects.
>
> **7) What is dk in Eq.2?**
>
> In the original paper of self attention[2], $d_{k}$ is used as the scaling factor, because when the dimensions of Q and K are large, the variance of their inner product will be large. In our experiment, $d_{k}$ is assigned to 1, scaling is not considered. We will add a description of it in the article.
>
> **About your constructive suggestions:**
>
> Thank you very much for your useful suggestions. We agree with you that using eye movement data as the label of significant areas is more convincing in cognition, which can also be the research direction of our future work.  We explained the reason why we use artificial annotation of foreground objects as foreground attention in paper ‘Our contributions’. Existing studies show that the human visual system is more inclined to focus on prominent objects, and the neural representation of these prominent objects in the brain is more obvious[3][4](Ungerleider & G, 2000; Braun et al., 2001) . The distribution of attention leads to an information bottleneck, only the most prominent objects are allowed to appear in the inferior temporal cortex, especially the ventral visual stream that encodes the identity of the object. The visual attention mechanism is crucial for simulating the neural response of the higher visual system[5][6] (Poggio & Anselmi, 2016; Khosla et al., 2020).
>
> [1] Tao Fang, Yu Qi, and Gang Pan. Reconstructing perceptive images from brain activity by shapesemantic gan. Advances in Neural Information Processing Systems, 33, 2020.
>
> [2] Ashish Vaswani, Noam Shazeer, Niki Parmar, Jakob Uszkoreit, Llion Jones, Aidan N Gomez,
> Łukasz Kaiser, and Illia Polosukhin. Attention is all you need. In Advances in neural information
> processing systems, pp. 5998–6008, 2017.
>
> [3] Sabine Kastner Ungerleider and Leslie G. Mechanisms of visual attention in the human cortex.
> Annual review of neuroscience, 23(1):315–341, 2000.
>
> [4] Jochen Braun, Christof Koch, Joel L Davis, et al. Visual attention and cortical circuits. MIT Press, 2001.
>
> [5] Tomaso Poggio and Fabio Anselmi. Visual cortex and deep networks: learning invariant representations. MIT Press, 2016.
>
> [6] Meenakshi Khosla, Gia Ngo, Keith Jamison, Amy Kuceyeski, and Mert Sabuncu. Neural encoding with visual attention. Advances in Neural Information Processing Systems, 33, 2020.

---

> ### Author Response · Authors · 2021-11-21
> **To Reviewer 48tB**
>
> We really appreciate your detailed review of our paper. Thank the reviewer for constructive suggestions and comments on our work, as well as affirmation of our innovations. Our replies are as follows:
> **About your main concern:**
>
> We are very sorry about that our presentation was not clear and caused your confusion. The reason why we design the F-attention decoder: we decode the F-attention corresponding to the image from fMRI data, and then apply the F-attention to guide Loop-Enc-Dec to complete the image reconstruction task. If we do not design such a decoder, we only have fMRI data and no information about the corresponding image and F-attention during the test. Therefore, in order to apply F-attention to guide the reconstruction task during the test, we need to input fMRI data into the trained decoder, and then obtain the significant regional distribution (F-attention) of the test image, next we complete the reconstruction task of the test image.
>
> **About your other concerns:**
>
> **1)The description of 'Visual attention labelling'.**
>
> We will rewrite this part in the paper to make the process clearer. The annotation process of our visual attention is a manual annotation process. In the first step, we manually intercept the foreground prominent object in the image, and then in the second step, we change the image into a single channel image, change the background to black (pixel value 0) and the foreground object to white (pixel value 255), The third step is to assign the attention weight. "Perform a binary value" means the background attention weight is 0 and the foreground object attention weight is 1. So far, the data annotation is completed.
>
> **2) the training loss process**
>
> We're sorry we didn't cover this part in the paper. In the process of training, the loss function of each part can converge well. *We will add the process of loss optimization in the training process in the appendix.*
>
> **3) the contribution of each part**
>
> As you mentioned in main concern, because of the shortcomings in our writing, you do not have a better understanding of the role of each part. We will optimize this defect in the modification of the paper to make the role of each part more obvious. For ablation experiments, we removed f-attention and lossloop from the complete framework, and then compared the roles of these two parts in the model. The results show that the experimental effect decreases after removing these two parts respectively, and the decline is more obvious after removing these two parts at the same time, which proves the effectiveness of our proposed f-attention and lossloop. We think the reduction of the parameters of the ablation experimental model is inevitable.
>
> **4) The choose of $\alpha$ and $\beta$ in Eq.4 (also $\lambda$ and $\mu$ in Eq.5) seems to be empirical**
>
> We tested the sequential combination parameters of  = 0.1, 0.3, 0.5, 0.7, 0.9 and  = 0.9, 0.7, 0.5, 0.3, 0.1 respectively, and selected the best combination,  $\lambda$=0.7 and $\mu$=0.3. It may not be the best among all the values of  and , but it is appropriate. We will supplement the details of this part in the appendix. $\alpha$ and $\beta$ goes through the same selection process.

---

### Official Review · Reviewer_UCCu · 2021-11-02

**Correctness:** 3
**Technical Novelty And Significance:** 2
**Empirical Novelty And Significance:** 2
**Recommendation:** 3
**Confidence:** 5

**Main Review:**

Strengths:
* Brain decoding is an interesting and still very much under-explored area
* Proposed a new framework by introducing the F-attention to enhance the image reconstruction further.

Weaknesses:

* The experimental section is weak, and hard to interpret the reconstruction results.
** Previous works had shown reconstruction results on at least two datasets. Why do authors consider only one fMRI natural image dataset of 1250 images? (Why not the BOLD 5000 fMRI dataset? And Vim-1 dataset)
** Authors can also report the 2-way, 5-ways, and 10-way results for better comparison with previous methods.
** From Figure 6, Without any statistical significance test on all the results, it is difficult to compare which method performs better than the remaining methods.
** Also, the proposed method has marginal improvement (68.9%) in terms of SSIM over the previous method (68.4%). No clear explanation for the comparison of the results.
** Figure 4 shows that ResNet is used as a pretrained encoder for extracting fMRI from stimuli, and what is the purpose of VGG19 in the Loop-Enc-Dec architecture? Throughout the paper, there is no discussion on VGG19.
** Any insights on self-attention?
** What is the dimension of Query, Key, and Value weight matrices?
** How is SSIM calculated?
** How do authors obtain λ value as 0.7 and μ as 0.3?
* Paper: M. Mozafari, L. Reddy and R. VanRullen, Reconstructing Natural Scenes from fMRI Patterns using BigBiGAN, this paper is not cited at all, and no comparison with previous works in terms of evaluation metrics.
* None of the image reconstruction results look comparable to Ground Truth or with previous methods.
* Cognitive insights are missing in this paper.
* Figure 8 illustrates that Lower visual areas have better reconstruction performance than Higher visual areas? Does this indicate the proposed model is reconstructing shapes and corners more? Is it because the number of voxels present in LVC is more than HVC?
* The total number of voxels present in VC?
* Among all the brain ROIs, which brain ROI is highly involved in image reconstruction?
* Using the Loop-Enc-Dec method to reconstruct natural images from fMRI is not new to this field.
* The mathematical notations used in Equations 5, 6, 7, 8, and 9 are difficult to follow.
** The naming conversion is bad.

**Summary Of The Paper:**

The paper proposed a Loop-Enc-Dec framework to perform an image reconstruction in a neural decoding task. The solution is based on an end-to-end encoder-decoder model under the guidance of Foreground-attention to enhance the perceptual quality of reconstructed images. The experimental results show visible improvements in the quality of reconstructed images without requiring additional fMRI training data. Also, the proposed method shows improvements in the structural similarity of image reconstruction.

**Summary Of The Review:**

Although the paper tests all of their contributions, the benefits do not seem to be dramatic. I do not see proper evaluation metrics that are used in previous methods for the model evaluation and also not tested on other existing fMRI datasets. The experimental section is weak, and it is hard to reproduce the results, so the paper does not introduce a significantly better technique.

---

> ### Author Response · Authors · 2021-11-21
> **To Reciewer UCCu**
>
>
> **2)Paper: M. Mozafari, L. Reddy and R. VanRullen, Reconstructing Natural Scenes from fMRI Patterns using BigBiGAN.**
>
> **·** Thank you very much for providing us with very useful related work, we will cite it and compare with it in the later article modification. In good faith, we have read this article carefully before. Its research task is to reconstruct images of the same category as visual stimulation images from fMRI data, rather than visual stimulation images themselves. So we simply think there is a lack of comparability with our work, and we didn't quote it in the article at the beginning, but we will compare with it in the later article modification.
>
> **3)None of the image reconstruction results look comparable to Ground Truth or with previous methods.**
>
> **·** We understand your concern, our experimental results are not significantly improved compared with the previous work. In good faith, in previous works, the intuitive feeling of reconstructed images is not much different. For example, Fang et al.[1], the visual clarity of images has not been greatly improved compared with Beliy et al.[2], it has made some improvement in the quantitative Pairwise SSIM index. In this work, authors will focus on the innovation of methods. We focus on how to propose an innovative method to reconstruct the visual stimulation images. As far as we know, we introduce the saliency map (Foreground-attention) of the image into this work for the first time, and successfully decode the saliency region distribution (Foreground-attention) of the human viewing image from the fMRI data. Then we use the Foreground-attention to guide the image reconstruction model Loop-Enc-Dec, and reconstruct visual stimulus images successfully. At the same time, our results have made some improvement in quantitative evaluation.
>
> **4)Cognitive insights are missing in this paper.**
>
> **·** We consider that the human brain is strongly wound into sulci and gyri, some spatially adjacent voxels are not connected in practice, so it is necessary to consider the global information when decoding fMRI. We introduce the self-attention module for capturing global information into the process of decoding F-attention.
>
> **5)Figure 8 illustrates that Lower visual areas have better reconstruction performance than Higher visual areas?**
>
> **·**  We are sorry for the inadequacy of our work. Fang et al. discussed similar problems in this paper[1]. They think that the shape information reconstructed from the low-level visual cortex gets a higher score on pairwise SSIM because the low-level visual cortex can better capture shape information. We compared the reconstruction results of LVC and HVC, and found that the low-level visual cortex can intuitively better retain the shape and direction information, and obtain a higher score on pairwise SSIM. Therefore, we think that the low-level visual cortex may better capture the shape information.
>
> **6)The total number of voxels present in VC?**
>
> **·**  The total number of voxels present in VC is 4643.
>
> **7)Among all the brain ROIs, which brain ROI is highly involved in image reconstruction?**
>
> **·** We use VC ROI to reconstruct the visual stimulus images.
>
> **8)Using the Loop-Enc-Dec method to reconstruct natural images from fMRI is not new to this field.**
>
> **·** Enc-Dec method is widely used to reconstruct natural images from fMRI. In this paper, we improve the loss function of ENC-Dec and add $Loss_{Loop}$ to increase the constraints in the training process. Ablation experiments show that our improvement is effective.
>
> **9)The mathematical notations used in Equations 5, 6, 7, 8, and 9 are difficult to follow.**
>
> **·** Thank you very much for your useful suggestion. We will simplify the naming of formulas and variables in the modification of the article.
>
> [1] Tao Fang, Yu Qi, and Gang Pan. Reconstructing perceptive images from brain activity by shapesemantic gan. Advances in Neural Information Processing Systems, 33, 2020.
>
> [2] Roman Beliy, Guy Gaziv, Assaf Hoogi, Francesca Strappini, Tal Golan, and Michal Irani. From
> voxels to pixels and back: Self-supervision in natural-image reconstruction from fmri. Advances in Neural Information Processing Systems, 32:6517–6527, 2019.
>
> [3] Guohua Shen, Kshitij Dwivedi, Kei Majima, Tomoyasu Horikawa, and Yukiyasu Kamitani. Endto-end deep image reconstruction from human brain activity. Frontiers in Computational Neuroscience, 13:21, 2019a.

---

> ### Author Response · Authors · 2021-11-21
> **To Reviewer UCCu**
>
> We really appreciate your detailed review of our paper. Thank you for constructive suggestions and comments on our work, as well as affirmation of our innovations. Our replies are as follows:
>
> **1)The experimental section is weak, and hard to interpret the reconstruction results.**
>
> **·** We will experiment on other data sets later. In many papers, we see that the authors only experiment on this data set. For example, our experiment is compared with the experiment of Fang et al.[1]. In this paper, the authors only use the data set of 1250 images. Therefore, we only conducted experiments on this data set at the beginning.
>
> **·** In order to compare with experimental results of Fang et al.[1], we used the comparison method of 2-way Pairwise SSIM which is used in Fang et al.[1]. At the same time, we note that this paper (Fang et al.[1]) was published in 2020, which is later than Beliy et al.[2] and Shen et al.[3], but the 2-way pairwise SSIM reported by it decreased significantly compared with the original reports in Beliy et al.[2] and Shen et al.[3]. Therefore, we think that the results in Beliy et al.[2] and Shen et al.[3] are not comparable, so we only use the comparison method of 2-way pairwise SSIM to compare with the results reported in Fang et al.[1].
>
> **·** We are sorry for your question due to our unclear explanation. In Figure 6, we show some experimental results, and conduct quantitative evaluation on all test sets. The quantitative evaluation value is the average result of five experiments. We have rewritten this part.
>
> **·** Our experimental results achieved 68.9% on 2-way pairwise SSIM, slightly better than 68.4%, indicating the effectiveness of our method and foreground-attention can guide loop-Enc-Dec to complete the reconstruction of visual stimulation images.
>
> **·** We are sorry for your question due to our unclear writing. We mentioned the function of VGG19 in 2.3.2. We have rewritten this part to ensure that the logic of the text is clear. VGG19 is used as a feature extractor to extract the convolution features of visual stimulation images and reconstructed images, and then calculate the loss between convolution features to obtain $loss_{image}$ and $loss_{imageNet}$.
>
> **·** Self-attention is the basis of transformer model and was first applied in NLP field [4]. Its function is to capture the global information of features and solve the problem of long-distance dependence of common models LSTM and RNN in NLP field. Transformer has been successfully applied to CV field in recent years. Because the human brain is strongly wound into sulci and gyri, some spatially adjacent voxels are not connected in practice. Therefore, it is necessary to consider the global information when decoding fMRI, so we introduce the self-attention module for capturing global information into the process of decoding F-attention.
>
> **·** The dimension of Query and Key is [68, 68, 8], and the dimension of Value is [68, 68, 64].
>
> **·** SSIM is an index to measure the similarity of two images[5], the calculation formula is as follows:
>
> $$
>  SSIM = \frac{(2\mu_{x}\mu_{y}+c_{1})(2\sigma_{xy}+c_{2})}{(\mu_{x}^{2}+\mu_{y}^{2}+c_{1})(\sigma_{x}^{2}+\sigma_{y}^{2}+c_{2})}
> $$
>
> Where $x$ and $y$ are two images, $\mu_{x}$ is the mean of $x$, $\mu_{y}$ is the mean of $y$, $\sigma_{x}^{2}$ is the variance of $x$, $\sigma_{y}^{2}$ is the variance of $y$, $\sigma_{xy}$ is the covariance of $x$ and $y$,  $c_{1}=(k_{1}L)^2$, $c2=(k_{2}L)^2$, is a constant used to maintain stability. $L$ is the dynamic range of pixel values. $k_{1}$=0.01, $k_{2}$=0.03.
> *We will supplement the details of this part in the appendix.*
>
> **·** We tested the sequential combination parameters of $\mu$ = 0.1, 0.3, 0.5, 0.7, 0.9 and $\lambda$ = 0.9, 0.7, 0.5, 0.3, 0.1 respectively, and selected the best combination, $\lambda$=0.7 and $\mu$=0.3. It may not be the best among all the values of $\lambda$ and $\mu$, but it is appropriate. *We will supplement the details of this part in the appendix.*

---

### Official Review · Reviewer_Pz1R · 2021-11-03

**Correctness:** 3
**Technical Novelty And Significance:** 2
**Empirical Novelty And Significance:** 1
**Recommendation:** 3
**Confidence:** 3

**Main Review:**

Strength:
The encoder decoder loop and use of foreground is relatively novel.

Weakness:
It is not very convincing to me that the loop-enc-dec model and the use of foreground attention is able to reconstruct stimuli images from brain responses better than past attempts. The one (if not only) quantitive comparison the paper makes with past results (as in Figure 6) shows that this method barely has any improvement over other methods. In terms of qualitative comparison, it is not clear to me the reconstruction provided by this methods is better than those generated by other methods.

Other than that, this paper would benefit a lot from a more thorough editing - some wordings are very confusing and there are a lot of typos. For example, "deep learing models" in the second paragraph of the introduction. And in the results section (3.2), 68.4% should be from Beliy et. al, not Fang et. al.



**Summary Of The Paper:**

This paper proposes a novel method incorporating decoded foreground attention and a new training scheme with encoder and decode in a loop to reconstruct image stimuli from fMRI data.

**Summary Of The Review:**

This paper has some interesting and innovative ideas about recounting visual stimuli from brain responses. However the paper is not very well written, hard to understand and the results are not very convincing.

---

> ### Author Response · Authors · 2021-11-21
> **To Reviewer Pz1R**
>
> Thank you for constructive suggestions and comments on our work, as well as affirmation of our innovations. Our replies are as follows:
>
> In this paper, we propose an algorithm for reconstructing visual stimulation image based on foreground attention. The foreground attention of the image is successfully decoded from fMRI data, and then we use the foreground attention to guide the image reconstruction model Loop-Enc-Dec.
>
> **1)The experimental effect is not significantly improved compared with the previous work.**
>
> We understand your concern, our experimental results are not significantly improved compared with the previous work, we will improve our experimental results in the future work. In good faith, in previous works, the intuitive feeling of reconstructed images is not much different. For example, Fang et al.[1], the visual clarity of images has not been greatly improved compared with Beliy et al.[2], it has made some improvement in the quantitative Pairwise SSIM index. *In this work, authors will focus on the innovation of methods.* We focus on how to propose an innovative method to reconstruct the visual stimulation images. As far as we know, we introduce the saliency map (Foreground-attention) of the image into this work for the first time, and successfully decode the saliency region distribution (Foreground-attention) of the human viewing image from the fMRI data. Then we use the Foreground-attention to guide the image reconstruction model Loop-Enc-Dec, and reconstruct visual stimulus images successfully. At the same time, our results have made some improvement in quantitative evaluation.
>
> **2)Writing about the article.**
>
> Thank you for pointing out the defects. We will carefully check the writing of the article and correct the mistakes at the same time.
>
>
>
> [1] Tao Fang, Yu Qi, and Gang Pan. Reconstructing perceptive images from brain activity by shapesemantic gan. Advances in Neural Information Processing Systems, 33, 2020.
>
> [2] Roman Beliy, Guy Gaziv, Assaf Hoogi, Francesca Strappini, Tal Golan, and Michal Irani. From voxels to pixels and back: Self-supervision in natural-image reconstruction from fmri. Advances in Neural Information Processing Systems, 32:6517–6527, 2019.

---

### Decision · Program_Chairs · 2022-01-20

**Decision:**

Reject

**Comment:**

This paper present a model for reconstructing images from fMRI recordings, based on an encoder and decoder used in a loop. The reviewers were unanimous in their opinion that this paper is not ready for publication at this stage. They raised concerns ranging from the quality of the result and how to compare them to previous methods, to the justification behind different modeling choices. The authors were gracious in their responses to the reviewers. I do not recommend acceptance at this stage,